# NeuralTouch: Leveraging Implicit Neural Descriptor for Precise Sim-to-Real Tactile Robot Control

Yijiong Lin, Bowen Deng, Chenghua Lu, Max Yang, John Lloyd, Efi Psomopoulou, Nathan F. Lepora

## I. INTRODUCTION

Current applications of vision and touch for robotic manipulation tend to be constrained by several factors. First, in common scenarios objects are already optimally positioned in the hand for grasping by the robot [1], [2]. Second, policies can be restricted to manipulating objects or contact features that were trained already, and so lack the ability to generalize to novel objects [3], [4].Third, the independent use of vision and touch modalities can reduce their synergistic potential [5], [6]. Fourth, multimodal policies developed in simulation struggle to transition seamlessly to real-world environments. This paper seeks to address these challenges by proposing a novel multimodal policy learning framework capable of overcoming these limitations.

In this work, we present *NeuralTouch*, a tactile RL policy learning framework with neural descriptor fields (NDF) [7]. Our goal is to improve the grasping accuracy of NDF-based methods with touch while maintaining sufficient generalizability to different inter-category objects. Furthermore, this framework does not restrict the NDF-based tactile servoing to limited, predefined contact geometries.

Experimentally, we focus on precise grasping with a tactile gripper through the aforementioned visual (coarse) and tactile (fine) phases. Specifically, in the coarse phase, we use an NDF to generate a pre-grasping pose, then the fine phase focuses on in-hand tactile servoing of the gripper fingers that repositions and reorients the gripper to achieve a specific grasp. This process is challenging due to the need to interpret the underlying object geometry in combination with precise control of a 6-DoF robot arm and parallel jaw gripper.

The main contributions of this work are as follows:
1) We propose a deep-RL-based framework with neural descriptor fields to train a general tactile policy which does not need any explicit assumption about prior contact geometry.
2) We demonstrate that our NeuralTouch strongly complements state-of-the-art vision-based grasping to achieve the desired grasping pose with improved accuracy.
3) We validate this experimentally with zero-shot sim-to-real policy transfer and few-shot demonstration to showcase that our method solves a variety of downstream manipulation tasks over a variety of objects.

## II. METHODS

We separate the robotic grasping task into two phases: a coarse vision-guided phase and a fine tactile-guided phase. In

All authors are with the Department of Engineering Mathematics and Bristol Robotics Laboratory, University of Bristol, Bristol BS8 1UB, U.K. (email: {yijiong.lin, n.lepora}@bristol.ac.uk)

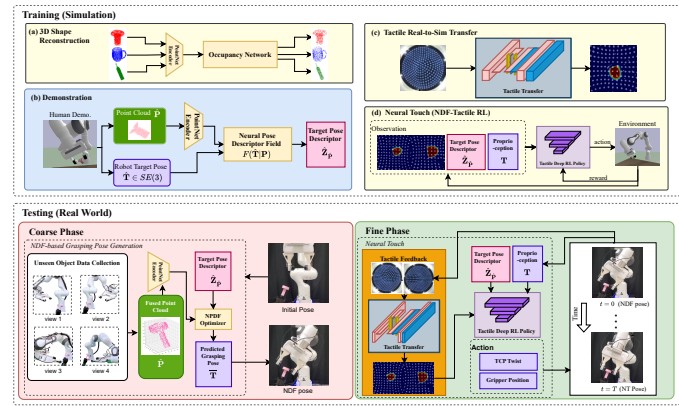

Figure 1. Overview of the NeuralTouch: In simulation, we first pre-train an occupancy network which is the core component of the Neural Pose Descriptor Fields. Secondly, we collect human demonstrations along with object point clouds and robot target grasping pose descriptors depending on the manipulation tasks. Thirdly, we train an RL policy with tactile and proprioceptive feedback, to achieve fine grasping poses implicitly specified by these collected descriptors. After obtaining the NPDF and a well-trained policy, our system is directly deployed in the real world with a real-to-sim tactile transfer to accurately grasp unseen objects, executing manipulation tasks such as unplugging a bolt-like USB and inserting it into a socket.

the coarse phase, we leverage the descriptor generated from NDF to calculate the coarse target grasping pose. Then, in the fine phase, we apply a tactile RL policy to accurately grasp an object with a desired contact pose represented by the NDF descriptor.

Specifically, we focus on learning a tactile RL policy that can be generalized to different target contact poses for different objects or tasks with the help of implicit neural descriptors from NDF. The tactile RL policy should not only consider the local contact to achieve safe, gentle contact but also have a sense of its desired contact pose with respect to the global shape of an object. Our method consists of three modules:
1) A PointNet Encoder [8] with Neural Pose Descriptor Fields [7] that learns implicit descriptors for various object shapes. These implicit representations describe the geometric relationships between poses (local frames) and the corresponding local shapes of inter-category objects.
2) A module to generate an initial coarse grasping pose using regression over the NDF [7].
3) A NeuralTouch RL module that learns a general tactile robotic policy conditioned on the implicit neural descriptors to achieve the desired fine grasping pose while maintaining safe, gentle physical interaction between the tactile robot and a manipulated object, given tactile and proprioceptive feedback.

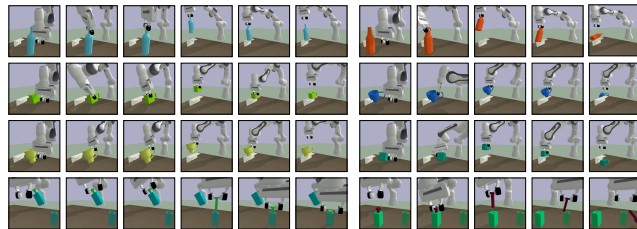

Figure 2. The snapshots of the robot performing four different tasks in simulation with three methods: (a) NeuralTouch, (b) NDF (first two rows) and NDF+RL-Touch (last two rows). From top to bottom row: object-pick-and-place (mug rim, mug horizontal handle, and bottle lid) and bolt-out/in-hole.

## III. Experiments and Results

### A. Tasks Setup

First, we design an ablation study and compare our method to two baselines: NDF [7] and NDF+RL-Touch. Specifically, we analyze grasping accuracy by measuring position errors and orientation errors for various target features of different objects in simulation. To further evaluate our proposed method, we consider two tasks both in simulation and in reality, with three phases: a) a coarse phase where the robot uses vision to locate and approach the unknown target feature pose of an object, b) a fine phase where the robot uses in-hand tactile servoing of the gripper fingers to achieve a precise grasping pose, and c) a replay phase where the robot executes a predefined skill to complete the task. The fine phase is particularly challenging due to the need for 7-DoF robot control and an understanding of the object's geometry to reposition and reorient the robot.

### B. Simulation Tasks Results

We evaluated the performance of the above methods in several fine manipulation tasks within simulation. Specifically, these are object pick-and-place and bolt-out/in-hole tasks, with 60 trials for each target feature. Our NeuralTouch method consistently outperforms the others in both tasks, because of its high accuracy.

### C. Real-world Tasks Results

In the bottle-lid-opening task, our NeuralTouch method achieves $90\%$ success rate for the bottles of apple juice and the ketchup, and achieves $85\%$ for the syrup bottle. In comparison, the vanilla NDF only achieved success rates of $30$–$45\%$. Thus, without tactile feedback, the NDF method frequently fails to open the bottle lid, as the rotation action must be executed precisely around the central axis of the cylinder-shaped lid to be successful. Also, when the gripper approached the lid with a large positional offset (where one finger was much closer to the lid), the lid would oscillate forwards then back rather than continuously turn. These behaviours are shown in the supplementary video.

In the peg-out/in-hole task, our NeuralTouch method achieved success rates of $55\%$, $25\%$ and $15\%$ for the bolt, plug and USB objects, respectively, consistent with the clearances of these objects progressively decreasing. Note that even

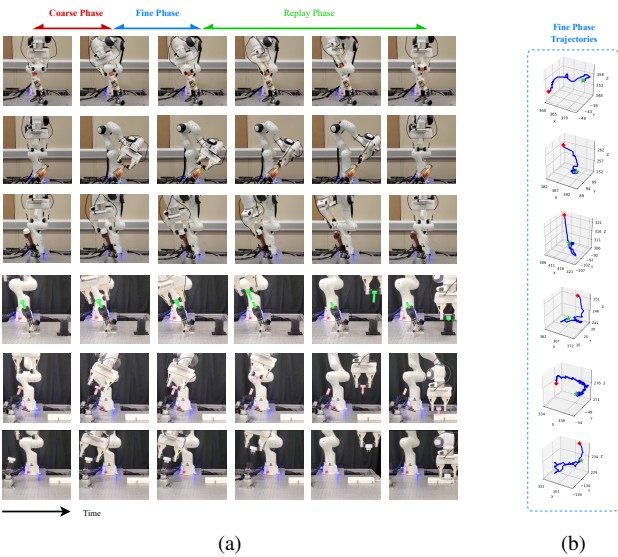

(a)        (b)

Figure 3. (a) Robot arm equipped with a tactile gripper performing two real-world manipulation tasks requiring high accuracy. Top 3 rows: bottle-lid opening. 4th row: peg-in/out-hole insertion. Bottom 2 rows: to increase the difficulty of the insertion task, we also experimented with a USB-head bolt and a plug where the clearances were approximately 0.5 mm and 1 mm, respectively. (b) End-effector trajectories recorded during the second phase. The red diamond represents the initial position determined by the NDF, while the green star indicates the final position achieved after tactile servoing.

though the success rates of NeuralTouch with the plug and the USB are not high, it does succeed sometimes, and this is a task in which the actions can be repeated until it succeeds. Therefore, another way to interpret the results is that they take a longer times to complete. Also, even when the task fails on the insertion, it only has about 1 mm error, compared to clearances of 1 mm and 0.5 mm respectively.

## IV. Conclusion

We presented NeuralTouch, a new method to achieve accurate robotic grasping that integrates vision and touch to enable precise manipulation with various objects and target features of those objects. Our approach consists of two main phases: a coarse phase, where the NDF is used to generate an initial grasping pose; and a fine phase, where the robot engages in tactile servoing using a neural descriptor-based RL tactile policy upon approaching the initial pose. Additionally, we demonstrate applications of our method by introducing a third replay phase, where the robot performs downstream tasks requiring high precision, such as peg-out/in-hole. Our ablation study shows that NeuralTouch significantly outperforms baseline methods in grasping accuracy and generalizability. Furthermore, our method is sim-to-real transferable, which makes it easy to deploy in real-world scenarios. In the future, exploring using tactile skin for large area contact with NeuralTouch will be an interesting direction [9].

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
