# OpenReview forum: "NeuralTouch: Leveraging Implicit Neural Descriptor for Precise Sim-to-Real Tactile Robot Control"
_IEEE.org/IROS/2025/Workshop/Tactile_Sensing — IROS 2025 Workshop Tactile Sensing OralPoster_

### Official Review · Reviewer_mNib · 2025-09-22
**Sim-to-Real tactile control**

**Rating:** 7
**Confidence:** 4

**Review:**

The manuscript introduces a tactile RL policy framework that leverages an initial vision coarse phase and a fine tactile phase. The vision provides global information, and the tactile provides local adjustment.

One question is that, besides the proprioceptive info of the robot, should the RL policy also take the position/pose of the object within the global frame at each time step as the input? The authors are advised to give information on that.

---

### Official Review · Reviewer_W8Jh · 2025-09-24
**Improve structure and expand evaluation**

**Rating:** 8
**Confidence:** 4

**Review:**

This paper introduces a promising multi-modal policy learning framework using NDFs, showing better grasping accuracy than the vision-only baseline. I recommend two main improvements. First, for better structure, please move the methodology description out of the introduction and into its own "Methods" section, starting from the fourth paragraph. Second, to strengthen the evaluation, please add more experimental results and a quantitative comparison to state-of-the-art (SOTA) methods. You have ample space to incorporate these changes and expand your work to the full two-page limit.